# Development of a definition and rules for causal attribution of post-colonoscopy bleeding

Robert J. Hilsden[1,2]*, Courtney M. Maxwell[3], Nauzer Forbes[1,2], Ronald J. Bridges[1], Alaa Rostom[4], Catherine Dube[4], Devon Boyne[2], Darren Brenner[2,5], Steven J. Heitman[1,2]

1 Department of Medicine, Cumming School of Medicine, University of Calgary, Calgary, AB, Canada, 2 Department of Community Health Sciences, Cumming School of Medicine, University of Calgary, Calgary, AB, Canada, 3 Forzani & MacPhail Colon Cancer Screening Centre, Alberta Health Services, Calgary, AB, Canada, 4 Department of Medicine, University of Ottawa, Ottawa, ON, Canada, 5 Department of Oncology, Cumming School of Medicine, University of Calgary, Calgary, AB, Canada

* rhilsden@ucalgary.ca

## Abstract

### Background

Post-colonoscopy bleeding (PCB) is an important colonoscopy quality indicator that is recommended to be routinely collected by colorectal cancer screening programs and endoscopy quality improvement programs. We created a standardized and reliable definition of PCB and set of rules for attributing the relatedness of PCB to a colonoscopy.

### Methods

PCB events were identified from colonoscopies performed at the Forzani & MacPhail Colon Cancer Screening Centre. Existing definitions and relatedness rules for PCB were reviewed by the authors and a draft definition and set of rules was created. The definition and rules were revised after initial testing was performed using a set of 15 bleeding events. Information available for each event included the original endoscopy report and data abstracted from the emergency or inpatient record by a trained research assistant. A validation set of 32 bleeding events were then reviewed to assess their interrater reliability by having three endoscopists and one research assistant complete independent reviews and three endoscopists complete a consensus review. The Kappa statistic was used to measure interrater reliability.

### Results

The panel classified 28 of 32 events as meeting the definition of PCB and rated 7, 8 and 6 events as definitely, probably and possibly related to the colonoscopy, respectively. The Kappa for the definition of PCB for the three independent reviews was 0.82 (substantial agreement). The Kappa for the attribution of the PCB to the colonoscopy by the three endosocopists was 0.74 (substantial agreement). The research assistant had a high agreement with the panel for both the definition (100% agreement) and application of the causal criteria (kappa 0.95).

**Data Availability Statement:** All datafiles are available from the Harvard Dataverse (Hilsden, Robert, 2019, "Attribution rules for post-

colonoscopy bleeding", https://doi.org/10.7910/DVN/KXNYAS, Harvard Dataverse).

**Funding:** The authors received no specific funding for this work.

**Competing interests:** The authors have declared that no competing interests exist.

## Conclusions

A standardized definition of PCB and attribution rules achieved high interrater reliability by endoscopists and a non-endoscopist and provides a template of required data for event adjudication by screening and quality improvement programs.

## Introduction

As organized screening for colorectal cancer has become commonplace, there has been a growing emphasis on monitoring the quality of colonoscopy. One critical indicator of colonoscopy quality is the rate of serious adverse events, such as bleeding and perforation. A recent meta-analysis reported the rate of bleeding based on 15 population-level studies to be 2.4 per 1,000 colonoscopies. [1–3] However, the lack of standardized and reliable definitions of post-colonoscopy bleeding and causal attribution rules could prevent valid comparisons of rates across programs or research studies.

The purpose of this study was to create a standardized and reliable definition of post-colonoscopy bleeding and a set of rules for attributing the bleeding event to a colonoscopy that could be applied by a non-physician research assistant. The secondary purpose was to create a set of core data elements required for assessing post-colonoscopy bleeding.

## Materials and methods

### Study design and patients

The study was conducted at the Alberta Health Services' Forzani & MacPhail Colon Cancer Screening Centre (CCSC) in Calgary, AB, Canada. The Centre is a publicly-funded endoscopy unit that only provides colonoscopies related to screening for colorectal cancer, including primary screening, post-polypectomy surveillance and diagnostic colonoscopy following a positive fecal immunochemical test. Colonoscopies performed for other indications, such as the investigation of signs or symptoms of gastrointestinal disease, are performed at hospital endoscopy units. All patients must be free of medical conditions that would place them at higher risk for colonoscopy-related adverse events (ASA Class I/II & select Class III). All patients undergo a consultation appointment with a trained nurse. Those who do not meet the Centre's eligibility criteria are redirected elsewhere. Colonoscopies at the Centre are performed by gastroenterologists and colorectal surgeons who also perform endoscopies at hospital endoscopy units.

The Centre maintains a quality improvement program that includes the identification of all emergency room visits and inpatient hospitalizations occurring within 30 days of a colonoscopy. Events are identified by linking a patient's unique personal health care number to the provincial ambulatory care database (emergency room visits) and discharge abstract database (inpatient stays). Additional information on all events is obtained by a review of the emergency room and/or inpatient chart by a trained research assistant.

For this study, we used post-colonoscopy bleeding events that occurred after colonoscopies performed from 2008 to 2014. Two sets of bleeding events were used in this study. The cases included in the two sets were selected to represent a range of bleeding events (clearly related to clearly unrelated). Information for each case included the anonymized index colonoscopy report, second colonoscopy (procedure performed to evaluate bleeding) report, and data abstracted from the emergency and inpatient charts.

This study received IRB approval by the Health Research Ethics Board of Alberta (HREBA. CC-18-0702), which granted a waiver from the requirement to obtain written informed consent. All data was originally collected by the CCSC's quality improvement program. Two members of the study team (RJH, CMM) had access to identifiable data, whereas the remaining study team only had access to anonymized patient information.

### Creation of post-colonoscopy bleeding definition and attribution rules

We created an initial definition of post-colonoscopy bleeding after reviewing published clinical practice guidelines. Next we created a set of rules for the causal attribution of the bleeding event to the colonoscopy using a framework commonly employed in clinical trials. [4] All authors then reviewed the first set of 15 bleeding events using the definition and attribution rules. A panel meeting was then held where each bleeding case was reviewed and the ease of use and appropriateness of the rules were discussed. The draft versions were then modified to create the final post-colonoscopy bleeding definition and attribution rules.

### Validation testing

The final post-colonoscopy bleeding definition and attribution rules were then evaluated in a second set of 32 bleeding events. Three endoscopists (RJH, NF, SH) as a panel reviewed each event to agree upon a consensus rating as to whether the event met the definition of post-colonoscopy bleeding and, if it did, how likely the event was due to the colonoscopy. They did twice, first using the criteria and second based on their overall global assessment. The panel's global assessment was defined as the "gold standard." Three other endoscopists (RB, CD, AR) reviewed and rated each event independently using the definition and attribution rules. Finally, a non-physician research assistant (CMM) experienced in abstracting adverse event data from colonoscopy and hospital records, reviewed and rated each event using the definition and attribution rules.

### Statistical analysis

The causal attribution variable was ordinal with five possible responses: definitely related, probably related, possibly related, unlikely related or unrelated (Table 1). A binary causal attribution variable was also created by classifying cases as related to the colonoscopy (definitely, probably or possibly related) or unrelated (unlikely related or unrelated).

Kappa statistics were used to assess the inter-rater reliability of the definition and causal attribution rules. The Kappa statistic quantifying the degree of agreement between the three clinicians, with respect to whether or not the event met the definition of a post-colonoscopy bleed, was estimated using the Nelson-Edward method. [5] The weighted Kappa statistic was used for the assessment of the causal attribution rules, as it is a five-point ordinal scale where a disagreement of one level between reviewers is less concerning than a disagreement of greater than one level. The model-based approach developed by Nelson and Edwards was used with quadratic weights to estimate the weighted Kappa statistic. [6] For both the binary and ordinal outcomes, we chose to use the Nelson-Edwards model-based approach because this method has been shown to provide more accurate estimates of inter-rater reliability that tend to be more conservative when compared to other methods. [6, 7] In addition, this model-based method can incorporate data from subjects who were not rated by one of the three clinicians, unlike other methods. [7]

The model-based kappa statistics were run using the *modelkappa* function (available at https://github.com/AyaMitani/modelkappa/) in R (R Core Team, Vienna, Austria). Descriptive statistics and estimation of Kappa statistics for causal attribution as a binary (related/unrelated) variable were performed using Stata version 15 (StataCorp LLC, College Station, Tx).

**Table 1. Rules for attributing post-colonoscopy bleeding to the index colonoscopy.**

| Causal Attribution | Criteria |
|---|---|
| Definite | • Active bleeding or adherent clot identified at 2nd colonoscopy |
| Probable (must have 3) | • Within 14 days of colonoscopy |
| | • Hot snare polypectomy |
| | • Polypectomy site with visible vessel at 2nd colonoscopy |
| | • High risk polypectomy[1] |
| Possible (must have 2) | • Within 14 days of colonoscopy |
| | • Hot snare polypectomy |
| | • Polypectomy site with visible vessel at 2nd colonoscopy |
| | • High risk polypectomy[1] |
| Unlikely (must have 3) | • 14 days after colonoscopy |
| | • Biopsy, cold snare polypectomy or no polypectomy |
| | • Low risk polypectomy |
| | • No high risk stigmata[2] or no 2nd colonoscopy |
| Unrelated | • Alternative source of bleeding identified |
| | • (Includes preparation-related Mallory-Weiss tears) |

[1] High risk polypectomy (any one of the following)

1. Size > 2cm.
2. Size 1–2 cm plus any one of
   • Location: Proximal to the transverse colon.
   • Shape: Pedunculated.
   • Patient uses anti-platelet therapy or oral anticoagulant.
3. Immediate post-polypectomy bleeding occurred that required intervention.

[2] High risk Stigmata: Active bleeding, Adherent clot, Visible vessel.

## Results

Post-colonoscopy bleeding was defined as "patient or health care provider reported rectal bleeding (other than blood on the toilet paper) and/or hemoglobin drop > 2 g/l within 30 days of a colonoscopy resulting in an emergency/urgent care center visit or hospital admission." The final attribution rules are shown in Table 1.

The consensus panel classified 28 of 32 events as meeting the definition of post-colonoscopy bleeding. The four cases that did not meet the definition were upper gastrointestinal hemorrhages where the primary presenting symptoms included hematemesis. Of the 28 cases that met the definition of post-colonoscopy bleeding, the panel's global assessment of causality classified 8, 13 and 1 event as definitely, probably and possibly related to the colonoscopy, respectively. The panel's assessment was that 5 events were unlikely to be due to the colonoscopy and 1 event was unrelated to the colonoscopy. Therefore, when rating events as due to the colonoscopy (definite, probably, possible) or not due to the colonoscopy (unlikely, unrelated), 22 events were rated as due to the colonoscopy and 6 events were rated as not due to the colonoscopy.

When using the causal criteria, the panel's consensus rating was 7, 8 and 6 events as being definitely, probably and possibly related to the colonoscopy, respectively. The panel rated 6 events as unlikely due to the colonoscopy and 1 event as unrelated to the colonoscopy.

The three independent reviewers classified 27, 27 and 29 events as meeting the definition of post-colonoscopy bleeding (kappa 0.82, 95% CI 0.59–1.0).

For the 32 events classified, the three independent reviewers had complete agreement on the classification of causality in 13 cases (unweighted Kappa 0.50, 95% CI 0.43–0.56). For only two of the cases did the reviewers disagree by more than one category (for example, definite

and possibly). The weighted Kappa was 0.74 (95% confidence interval 0.68–0.80) indicating substantial agreement. When restricting to the 28 classified by the panel as meeting the definition of post-colonoscopy bleeding, the unweighted and weighted Kappas were 0.42 (95% CI 0.34–0.49) and 0.67 (95% CI 0.59 to 0.75) respectively.

There was similarly high agreement among the reviewers when events were rated as either due to or not due to the colonoscopy (kappa 0.79, 95% CI 0.54–1.0). When considering the panel's global assessment as the gold standard, all three independent reviewers achieved a sensitivity of >95% and a specificity of 80%-100% for identifying a bleeding event due to a colonoscopy.

The research assistant was in complete agreement with the panel's classification of 28 of 32 events as meeting the definition of post-colonoscopy bleeding. When applying the causal criteria, the research assistant classified 7, 8 and 6 events as definitely, probably or possibly related to the colonoscopy, which was in agreement with the panel. The RA rated 5 events as unlikely related and 2 events as unrelated, therefore disagreeing with the panel on one event (kappa 0.95, 95% CI 0.85–1.0).

The data elements required for determining whether a bleeding event met the definition of post-colonoscopy bleeding and for determining its causal relationship to the colonoscopy are shown in Table 2. Missing information, such as polyp size, shape and method of removal and a precise description of the bleeding, was noted to be a problem by the panel reviewers and research assistant.

## Discussion

Post-colonoscopy bleeding, nearly always due to bleeding from a polypectomy site, is one of the procedure's most common complications. Therefore, it is recognized as an important colonoscopy quality indicator. Polypectomy site bleeding can be immediate (during the procedure) or delayed (after discharge from the endoscopy unit). Immediate bleeding often can be easily managed at the time of the endoscopy and therefore often results in no material impact on the clinical course of the patient.

**Table 2. Data elements required for each bleeding event.**

| Index Colonoscopy | | | |
|---|---|---|---|
| | A) Patient: | | |
| | | antiplatelet agents, anticoagulants (yes/no) | |
| | B) Immediate Post-Polypectomy Bleeding (yes/no) | | |
| | C)Polypectomy: | | |
| | | polypectomy with diathermy (hot) | |
| | | polypectomy with no diathermy (cold) | |
| | | no polypectomy | |
| | D) Polyp: | | |
| | | size (< 1 cm, 1–2 cm, > 2 cm) | |
| | | shape (pedunculated, not pedunculated) | |
| | | location proximal to transverse colon (yes/no) | |
| Bleeding Event | | | |
| | A) Time (days) since index colonoscopy: (< = 14 days, > 14 days) | | |
| | B) Description of bleeding | | |
| | C) Change in hemoglobin (if available) | | |
| | D) Second colonoscopy (to evaluate bleeding): | | |
| | | Done/Not done | |
| | | | If done: presence of active bleeding, adherent clot or visible vessel |

In 2008, the American Society for Gastrointestinal Endoscopy Quality Task Force convened a workshop on adverse events. The goals of this workshop included: (1) provide clear definitions for adverse events and (2) consider how to deal with delayed events, in particular, the issues of timing and attribution. Bleeding was defined as hematemesis and/or melena or hemoglobin drop > 2 grams. The report of the workshop recommended that adverse events be attributed to an endoscopic procedure using the ratings of definite, probable, possible or unlikely. However, no specific criteria were given to guide the use of these attributions.

In 2016, the British Society of Gastroenterology, the UK Joint Advisory Group on GI Endoscopy and the Association of Coloproctology of Great Britain and Ireland developed quality assurance measures specifically for colonoscopy. Post-polypectomy bleeding was defined as "rectal bleeding within 30 days of procedure resulting in any of the following": unplanned post-procedure medical consultation, unplanned hospital admission or prolongation of hospital stay, hemoglobin drop of $\geq 2$ g, transfusion, interventional procedure (endoscopic or radiological), surgery or death. The authors also recommended that "after root cause analysis, attribution of adverse events should be recorded as definite, probable, possible or unlikely." Again, no specific criteria were given to guide the use of these attributions.

Several problems can be faced when applying either the ASGE or UK recommendations in the setting of a colonoscopy screening program. First, the ASGE lexicon was created for use with all endoscopic procedures. The definition of bleeding includes hematemesis and melena (black stools), which would be unusual for post-polypectomy bleeding. Furthermore, in the screening setting, patients may not have a recent hemoglobin measurement and therefore it may not be possible to determine if a 2 g/L drop occurred. Finally, rectal bleeding can be trivial (trace blood on the toilet paper) and unlikely to be of clinical importance or related to post-polypectomy bleeding.

Our definition of post-colonoscopy bleeding is "patient or health care provider reported rectal bleeding (other than blood on the toilet paper) and/or hemoglobin drop > 2 g/l within 30 days of a colonoscopy resulting in an emergency/urgent care center visit or hospital admission." The key components of our definition includes: (1) we do not require objective medical evidence of bleeding by either a health care provider witnessing the bleeding or an objective drop in hemoglobin, (2) bleeding must be the passage of blood into the toilet and not just minor blood on the toilet paper, (3) we do accept an objective drop in hemoglobin to be adequate evidence of post-colonoscopy bleeding even if no rectal bleeding was noticed by the patient, and (4) bleeding must have resulted in the patient seeking urgent medical care. Our definition excludes immediate post-polypectomy bleeding that is controlled during the procedure and does not result in admission of the patient to hospital. It also excludes trivial or self-limited bleeding that neither impacts the patient nor generates a burden on the health care system.

We elected to exclude Mallory-Weiss tears, that can occur as a result of vomiting caused by the bowel preparation, from the events that would be included as a "Post-colonoscopy bleed." However, such bleeding events can either meet or not meet our definition of post-colonoscopy bleeding. For example, a tear that resulted in isolated hematemesis would not meet the definition, whereas a tear that resulted in only rectal bleeding would meet the definition. In our study, the panel classified the four cases of upper GI bleeding that all resulted in hematemesis as not meeting the definition of post-colonoscopy bleeding.

When developing criteria for attributing bleeding events to a colonoscopy, we considered published risk factors for post-polypectomy bleeding. For instance, we concluded that a bleeding event would be more likely to be due to the index colonoscopy if a polypectomy with diathermy was performed than if polypectomy was performed by cold-snare without diathermy [8–10] or if no polypectomy was performed. [11] Other established predictors of post-

polypectomy bleeding include the removal of large polyps (especially those > 2 cm), [12–17] proximally located polyps [16, 18–21] or pedunculated polyps. [16] Polypectomy on patients on antiplatelet agents or anticoagulants (even if discontinued before the procedure) has also been reported as a predictor of delayed bleeding, [14, 21] as has the occurrence of observed intraprocedural bleeding. [13, 19] Finally, we accepted that a colonoscopy performed to investigate bleeding could either confirm (if active bleeding from a polypectomy site was seen) or exclude (if an alternative bleeding site was seen) post-polypectomy bleeding.

By developing our definition of post-colonoscopy bleeding and attribution rules, we also defined the core data set required to apply them. This would assist quality assurance programs in establishing data collection procedures and minimizing the data required. However, all reviewers experienced challenges in applying the criteria when data was missing from the colonoscopy report. At times, reviewers had to infer polyp size and method of removal from image's captured at the endoscopy. The research assistant was much less confident in making these inferences. Deficiencies in colonoscopy reports has been noted in several studies. [22] All of the polyp and polypectomy details required to be collected about the index colonoscopy are consistent with elements recommended by the Quality Assurance Task Group of the National Colorectal Cancer Roundtable to be part of a standard colonoscopy report. [23] Therefore, a high quality colonoscopy report should contain all information required to apply the criteria.

In summary, we have developed a definition of post-colonoscopy bleeding and a set of criteria for attributing the bleeding event to the colonoscopy based on information that should be routinely available in a colonoscopy report and emergency room/impatient chart that demonstrate high-interrater reliability and can be applied by a non-physician research assistant.

## Author Contributions

**Conceptualization:** Robert J. Hilsden, Nauzer Forbes, Ronald J. Bridges, Alaa Rostom, Catherine Dube, Steven J. Heitman.

**Data curation:** Robert J. Hilsden, Courtney M. Maxwell, Alaa Rostom.

**Formal analysis:** Robert J. Hilsden, Nauzer Forbes, Ronald J. Bridges, Alaa Rostom, Catherine Dube, Devon Boyne, Darren Brenner, Steven J. Heitman.

**Methodology:** Robert J. Hilsden, Darren Brenner.

**Project administration:** Courtney M. Maxwell.

**Writing – original draft:** Robert J. Hilsden, Catherine Dube.

**Writing – review & editing:** Robert J. Hilsden, Courtney M. Maxwell, Nauzer Forbes, Ronald J. Bridges, Alaa Rostom, Devon Boyne, Darren Brenner, Steven J. Heitman.

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
