## [Decision Letter · Decision Letter 0]

21 May 2020

PONE-D-19-35059

Development of a definition and rules for causal attribution of post-colonoscopy bleeding

PLOS ONE

Dear Dr. Hilsden,

Thank you for submitting your manuscript to PLOS ONE. After careful consideration, we feel that it has merit but does not fully meet PLOS ONE’s publication criteria as it currently stands. Therefore, we invite you to submit a revised version of the manuscript that addresses the points raised during the review process.

We look forward to receiving your revised manuscript.

Kind regards,

Roberto Coppola, MD, FACS

Academic Editor

PLOS ONE

Journal Requirements:

2. In the ethics statement in the manuscript and in the online submission form, please provide additional information about the patient records used in your retrospective study, including:

a) whether all data were fully anonymized before you accessed them;

b) the date range (month and year) during which patients' medical records were accessed;

c) the source of the medical records analyzed in this work (e.g. hospital, institution or medical center name).

If patients provided informed written consent to have data from their medical records used in research, please include this information.

Reviewers' comments:

Reviewer's Responses to Questions

**Comments to the Author**

1. Is the manuscript technically sound, and do the data support the conclusions?

Reviewer #1: Yes

Reviewer #2: Yes

2. Has the statistical analysis been performed appropriately and rigorously? 

Reviewer #1: N/A

Reviewer #2: Yes

3. Have the authors made all data underlying the findings in their manuscript fully available?

Reviewer #1: Yes

Reviewer #2: Yes

4. Is the manuscript presented in an intelligible fashion and written in standard English?

Reviewer #1: Yes

Reviewer #2: Yes

5. Review Comments to the Author

Reviewer #1: I appreciate the work done by the Authors. The topic is interesting and this paper could be useful in order to produce objective assessments regarding quality of colonoscopy. The methodology is accurate. Limitations have been discussed. I would only suggest making minor changes that can limit overlaps between introduction and discussion.

Reviewer #2: This is an interesting and useful paper proposing an objective evaluation of post-colonoscopy bleeding in a colorectal cancer screening setting.

I have no major criticisms.

The Discussion may be substantially shortened.

6. PLOS authors have the option to publish the peer review history of their article (what does this mean?). If published, this will include your full peer review and any attached files.

Reviewer #1: No

Reviewer #2: No

---

## [Author Response · Author response to Decision Letter 0]

18 Jun 2020

In preparing the revision, we have considered the comments of the two reviewers:

Reviewer #1: I appreciate the work done by the Authors. The topic is interesting and this paper could be useful in order to produce objective assessments regarding quality of colonoscopy. The methodology is accurate. Limitations have been discussed. I would only suggest making minor changes that can limit overlaps between introduction and discussion.

Reviewer #2: This is an interesting and useful paper proposing an objective evaluation of post-colonoscopy bleeding in a colorectal cancer screening setting.

I have no major criticisms.

The Discussion may be substantially shortened.

We have shortened the introduction to limit overlap between the introduction and the discussion. Because of these changes to the introduction, the discussion was kept largely intact.

We have added the requested additional details to the ethics section within the methods section. We were asked to provide (i) the date range during which patients’ medical records were accessed and (ii) the source of the medical records analyzed in this work. The data was originally collected by the institution’s quality improvement program. We were provided secondary use of this data for this study but did not directly access patients’ medical records. As the primary review of medical records was not performed as part of this research the requested data was not provided. The institution’s quality improvement program collected this data on an ongoing basis over the years 2008 - 2019 from all hospitals and urgent care centre’s in the province of Alberta and from the institution’s endoscopy report system. 

We hope our revisions satisfy the reviewers and hope that our manuscript is now acceptable for publication in PLOS ONE.

---

## [Editor Report · Decision Letter 1]

25 Jun 2020

Development of a definition and rules for causal attribution of post-colonoscopy bleeding

PONE-D-19-35059R1

Dear Dr. Hilsden,

We’re pleased to inform you that your manuscript has been judged scientifically suitable for publication and will be formally accepted for publication once it meets all outstanding technical requirements.

Kind regards,

Roberto Coppola, MD, FACS

Academic Editor

PLOS ONE

Additional Editor Comments (optional):

The Authors have entirely answered to the questions of the Reviewers.

This study is fascinating, and I am in favor of a publication in the Journal.
---

## [Editor Report · Acceptance letter]

8 Jul 2020

PONE-D-19-35059R1 

Development of a definition and rules for causal attribution of post-colonoscopy bleeding 

Dear Dr. Hilsden:

I'm pleased to inform you that your manuscript has been deemed suitable for publication in PLOS ONE. Congratulations! Your manuscript is now with our production department. 

Kind regards, 

on behalf of

Professor Roberto Coppola 

Academic Editor

PLOS ONE